# The Compound *(E)*-2-Cyano-*N*,3-diphenylacrylamide (JMPR-01): A Potential Drug for Treatment of Inflammatory Diseases

**DOI:** 10.3390/pharmaceutics14010188

**Published:** 2022-01-13

**Authors:** Pablo Rayff da Silva, Renan Fernandes do Espírito Santo, Camila de Oliveira Melo, Fábio Emanuel Pachú Cavalcante, Thássia Borges Costa, Yasmim Vilarim Barbosa, Yvnni M. S. de Medeiros e Silva, Natália Ferreira de Sousa, Cristiane Flora Villarreal, Ricardo Olímpio de Moura, Vanda Lucia dos Santos

**Affiliations:** 1Programa de Pós Graduação em Ciências Farmacêuticas, Universidade Estadual da Paraíba, Campina Grande 58429-500, PB, Brazil; pablo-rayff@hotmail.com (P.R.d.S.); camillamello-@hotmail.com (C.d.O.M.); ricardo.olimpiodemoura@gmail.com (R.O.d.M.); 2Laboratório de Ensaios Farmacológicos, Departamento de Farmácia, Universidade Estadual da Paraíba, Campina Grande 58429-500, PB, Brazil; fabiocavalcante221@gmail.com (F.E.P.C.); thassiacosta5@gmail.com (T.B.C.); yasmimvilarim.b@gmail.com (Y.V.B.); 3Laboratório de Desenvolvimento e Síntese de Fármacos, Departamento de Farmácia, Universidade Estadual da Paraíba, Campina Grande 58429-500, PB, Brazil; Yvnnim@gmail.com; 4Instituto Gonçalo Moniz, Fundação Osvaldo Cruz, Salvador 40296-710, BA, Brazil; r.fernandes88@hotmail.com (R.F.d.E.S.); cfv@ufba.br (C.F.V.); 5Faculdade de Farmácia, Universidade Federal da Bahia, Salvador 40170-290, BA, Brazil; 6Programa de Pós Graduação em Produtos Naturais, Sintéticos e Bioativos, Universidade Federal da Paraiba, João Pessoa 58051-900, PB, Brazil; nferreiradesousa.nfs@gmail.com

**Keywords:** bioisosterism, phenylacrylamide, immunomodulation, inflammation

## Abstract

The compound *(E)*-2-cyano-*N*,3-diphenylacrylamide (JMPR-01) was structurally developed using bioisosteric modifications of a hybrid prototype as formed from fragments of indomethacin and paracetamol. Initially, in vitro assays were performed to determine cell viability (in macrophage cultures), and its ability to modulate the synthesis of nitrite and cytokines (IL-1β and TNFα) in non-cytotoxic concentrations. In vivo, anti-inflammatory activity was explored using the CFA-induced paw edema and zymosan-induced peritonitis models. To investigate possible molecular targets, molecular docking was performed with the following crystallographic structures: LT-A4-H, PDE4B, COX-2, 5-LOX, and iNOS. As results, we observed a significant reduction in the production of nitrite and IL-1β at all concentrations used, and also for TNFα with JMPR-01 at 50 and 25 μM. The anti-edematogenic activity of JMPR-01 (100 mg/kg) was significant, reducing edema at 2–6 h, similar to the dexamethasone control. In induced peritonitis, JMPR-01 reduced leukocyte migration by 61.8, 68.5, and 90.5% at respective doses of 5, 10, and 50 mg/kg. In silico, JMPR-01 presented satisfactory coupling; mainly with LT-A4-H, PDE4B, and iNOS. These preliminary results demonstrate the strong potential of JMPR-01 to become a drug for the treatment of inflammatory diseases.

## 1. Introduction

Inflammation is the body’s reaction to aggressive chemical, physical, or biological agents, promoting coordinated activation of signaling pathways with the aim of instituting repair processes. The process is initiated by activation of standard cell surface receptors, which recognize stimuli, and activate intracellular signaling cascades, inducing the translocation of transcription factors and resulting in the expression of various inflammatory mediators [1,2]. At the tissue level, during the inflammatory response, typical acute symptoms such as edema, redness, heat, and pain, which may progress to loss of tissue function, are observed [3,4].

An acute process, if unresolved, can become both chronic and part of disease pathogenesis. According to the World Health Organization (WHO), inflammatory diseases are the third leading cause of death, accounting for about 3.46 million deaths, corresponding to 10.8% of the world total [5]. According to Global Business Intelligence Research, Non-steroidal anti-inflammatory drugs (NSAIDs) and steroids constitute one of the main classes of drugs used worldwide, mobilizing USD 85.9 billion in 2017 [6].

Many classes of drugs, such as non-steroidal anti-inflammatory drugs (NSAIDs) and corticosteroids have been shown to be effective in the treatment of inflammatory disorders [5,7]. However, when used for prolonged periods or in non-therapeutic doses, these agents can promote gastrointestinal, renal, and liver disorders, and in the case of steroids, immunosuppression and metabolic changes. Research for new clinically acceptable compounds and bioactive molecules against inflammation is therefore a continuing demand.

As for the development of promising drugs with anti-inflammatory potential, medicinal chemistry has become an important tool in planning new molecules with therapeutic potential, aimed at maintaining pharmacological response and with better toxicity profiles than previously described for conventional anti-inflammatory drugs. In this context, new structures are synthesized using privileged carbon skeletons as a basis, such as the phenylacrylamide function. Our research aimed to synthesize and investigate the anti-inflammatory potential of *(E)*-2-cyano-*N*,3-diphenylacrylamide, obtained by organic synthesis from bioisosteric modifications of a previously studied drug [8] and resulting from the hybridization of molecular fragments of the anti-inflammatory drugs indomethacin and paracetamol.

## 2. Materials and Methods

### 2.1. General Procedure for the Synthesis of Compound (E)-2-Cyano-N,3-diphenylacrylamide (JMPR-01)

To obtain the JMPR-01, the following products were used in equimolar (1 mmol): 2-cyano-N-phenylacetamide (JM-01), previously synthesized; and 2-carboxyaldehyde (Sigma-Aldrich, St. Louis, MO, USA). Toluene (10 mL) (Vetec, Rio de Janeiro, Brazil) was used as reaction medium, added with 5–10 drops of the catalyst, triethylamine (Sigma-Aldrich, St. Louis, MO, USA) (Figure 1). The reaction was kept under magnetic stirring, for 24 h at a temperature between 105–110 °C, and its end was observed by Analytical Thin Chromatography (CCDA). The reaction was filtered and the obtained crystals were washed with ice-cold distilled water, and then recrystallized in ethanol–water 1:1. The product was then analyzed by ^1^H and ^13^C NMR (Agilent NMR spectrometer, model Mercury Plus 500 MHz, OXFORD 300 magneto-NMR), infrared spectroscopy (IRPrestige-21 Spectrophotometer) and mass spectrometry (Shimadzu^®^ AXIMA series MALDI-TOF/MS).

### 2.2. Physical-Chemical Properties and Spectroscopic Data of (E)-2-Cyano-N,3-diphenylacrylamide (JMPR-01)

JMPR-01 was obtained as gray powder (72.37%). M.p. (°C): 200,26. Rf (7:3 AcOEt/n-hexane): 0.60. IR (ATR): 1596 (NH, amide folding), 1682 (C=O), 2227 (CN), 3317 (NH, secondary amide stretch), 3053 (=C-H), 1531–1440 (C=C, Ar) cm^−1^. NMR 1H (500 MHz, DMSOd6): 10.43 (1H, s, NH amide); 8.30 (1H, s, C=CH); 8.05–7.96 (2H, m, Ar phenylacetamide); 7.72–7.66 (2H, m, Ar phenylacetamide); 7.39 (2H, t, J = 7.9 Hz, Ar phenylacetamide) 7.67–7.57 (3H, m, Ar phenylacetamide); 7.19–7.12 (2H, m, Ar phenylacetamide). NMR de 13C (124 MHz, DMSOd6): δ 160.97 (C, C=O); 151.27 138,68 (C, N-Ar phenylacetamide), 132,94, (C, Ar phenylacetamide), 132,37 (C, Ar phenylacetamide), 130,52 (C, Ar phenylacetamide), 129,77 (C, Ar phenylacetamide) 129,24 (C, Ar phenylacetamide), 124,84 (C, Ar phenylacetamide), 121,02 (C, Ar phenylacetamide), 116,68 (C, CN), 107,81 (C, adjacent to CN). HRMS m/z [M + H] + calculated for C_16_ H_12_N_2_O; 248.09 found: 249.01.

### 2.3. Thermal Characterization

#### 2.3.1. Differential Scanning Calorimetry (DSC)

The Differential Scanning Calorimetry curve was obtained in a differential exploratory module of the Q20 (TA^®^—Instruments, New Castle, DE, USA). Samples of 2.00 ± 0.05 mg were used, placed in hermetically sealed aluminum crucibles, analyzed at a heating ratio of 10 °C min^−1^ with a temperature of 25 °C to 400 °C, under a nitrogen atmosphere with flow 50 mL min^−1^.

#### 2.3.2. Thermogravimetric (TG)

The thermogravimetric curve of JMPR-01 was obtained in a Pyris 1 TGA (Perkin Elmer^®^, Boston, MA, USA) thermal analyzer using alumina crucibles with 8 ± 0.1 mg samples, under nitrogen atmosphere at a 50 mL min^−1^ flow rate. The experiments were carried out in the temperature range of 25–900 °C, with a heating rate of 10 °C min^−1^.

### 2.4. Biological Activity

#### 2.4.1. Macrophages Cytotoxicity

The cytotoxicity of JMPR-01 was determined using the cell line J774 according to the methodology proposed by Mosmann (1983) [9], adapted according to Espirito–Santo (2017) [10]. Macrophages were seeded in 96-well plates (2 × 10^5^ cells/well) in Dulbecco’s Modified Eagle Medium (DMEM; Life Technologies^®^, Gibco BRL^®^, Gaithersburg, MD, USA), supplemented with 10% fetal bovine serum (Gibco BRL^®^, Gaithersburg, MD, USA) and 50 µg/mL of gentamicin (Novafarma, Anápolis, GO, Brazil). Afterwards, the plates were incubated for 2 h at 37° in an atmosphere of 5% CO_2_. The test molecule was added to the wells in triplicate at concentrations of 100, 50, 25, 12.5 and 6.25 µM, and incubated again for 72 h. As a positive control, 10 μM gentian violet (Synth, São Paulo, Brazil) was used. Finally, 20 µL/well of Alamarblue Cell Viability Reagent (Invitrogen^®^, Carlsbad, CA, USA) was added to the plate, holding for 12 h. Colorimetric scanning was performed at a length from 570 to 600 nm.

#### 2.4.2. Assessment of Cytokine and Nitric Oxide Production by Macrophages

To determine the influence of JMPR-01 on cytokines and nitric oxide (NO), J774 macrophages were seeded in 96-well culture plates at a concentration of 2 × 10^5^ cells per well in DMEM medium supplemented with 10% FBS and 50 μg/mL of gentamicin for 2 h at 37 °C under an atmosphere of 5% CO_2_. After seeding, the cells were treated with the molecule, at non-cytotoxic concentrations, with the vehicle (negative control) and with dexamethasone 40 µM (positive control), and then stimulated with LPS (500 ng/mL, Sigma, St. Louis, MO, USA) and IFN-γ (5 ng/mL; Sigma, St. Louis, MO, USA) and incubated at 37 °C. After an incubation period of 4 h (to measure TNF-α) and 24 h (to quantify IL-1β and nitrite), cell supernatants were collected, and kept at −80 °C. Then, cytokine concentrations were determined by enzyme-linked immunosorbent assay (ELISA), using the DuoSet kit from R & D Systems (Minneapolis, MN, USA). The quantification of nitric oxide was performed using the Griess method according to Green et al. (1982) [11].

#### 2.4.3. Animals

Adult male Swiss Webster mice weighing between 25 and 35 g, obtained from the Prof. Eduardo Barbosa Beserra Animal Facilities at the Paraíba State University and the Gonçalo Moniz Institute (FIOCRUZ; Salvador, Brazil), were used. In the vivarium, the animals were kept in plastic cages, under room temperature and humidity (23 ± 2 °C), with a 12 h light-dark cycle, and fed with feed and water ad libitum. All studies were performed between 08:00 and 17:00 p.m. The animal care and handling procedures were followed in strict accordance with the recommendations of the Guide for the Care and Use of Laboratory Animals of the National Institutes of Health and the Brazilian College of Animal Experimentation. Previously, the project was approved by the Ethics Committee on Animal Use under number 003/2021. All procedures from the beginning of the study until the time of euthanasia were performed to avoid suffering and reduce the discomfort and pain of the animals.

#### 2.4.4. Inflammatory Model

##### Zymosan-Induced Acute Peritonitis in Mice

Mice were divided into groups: positive control, negative control and test groups, which were treated orally with saline (5% DMSO), indomethacin 10 mg/kg and JMPR-01 at doses of 5, 10 and 50 mg/kg, respectively. After 1 h of the treatments, 0.25 mL of 2% zymosan was injected intraperitoneally. Four hours after induction of inflammation, the animals were euthanized by administering 2 mL of heparinized phosphate-buffered saline into the intraperitoneal cavity [12,13]. At the end, an incision was made, collecting the exudate, whose cells were resuspended in 500 μL of PBS and 10 μL of Turk’s fluid (1:20). To count the leukocytes, a Neubauer chamber was used under light microscopy, examining the four external quadrants.

##### Plesthismometer Test

Mice were lightly anesthetized with halothane and received an intraplantar injection of complete Freund’s adjuvant (CFA) (Sigma) in the right paw in a final volume of 20 μL, according to previously reported method [14]. JMPR-01 (50 and 100 mg/kg) or vehicle (5% DMSO in saline; control group) was administered by p.o. route 40 min before phlogiston agent. Dexamethasone (2 mg/kg, i.p) was used as standard. The paw volume was assessed (mm^3^) by plesthismometer (Ugo Basile, Comerio, Italy) as described above. The amount of paw swelling was determined for each mouse and data were represented as paw volume variation (Δ, mm^3^).

### 2.5. Docking Studies

The structure of the JMPR-01 was built with Chemdraw professional 3D 15.0 software, optimized by a minimization of the molecular energy using molecular mechanics (MM2) and then saved as MOL2 files. With the use of AutoDockTools-1.5.6, non-polar hydrogens were merged with the corresponding carbons, and then partial charges of atoms were calculated using the Gasteiger procedure implemented in the AutoDockTools package. Finally, the rotatable bonds of the ligands were defined, the structures were saved as pdbqt and used for docking studies.

The crystal structures of Inducible nitric oxide synthase (PDB ID: 3E7G), Phosphodiesterase 4B (PDB ID: 1XMU), Leukotriene A4 hydrolase (PDB ID: 1HS6), 5-Lipoxygenase (PDB ID: 6NCF) and Cyclooxygenase-2 (PDB ID: 3LN1) were retrieved from RCSB Protein Data Bank as described [15,16]. With the use of Molecular Graphics System, PyMOL, water molecules and other heteroatoms were removed. Then, using AutoDockTools, non-polar hydrogens were merged, and polar hydrogens added to the structures of the proteins. Kollman charges were added and the structures were saved as pdbqt for the docking studies.

The Lamarckian genetic algorithm in AutoDock 4.2.6 was applied to search the best conformation and orientation of the ligands. The global optimization was started with a population of 150 randomly positioned individuals with a maximum of 2,500,000 energy evaluations and a maximum of 27,000 generations. During each docking experiment, 100 runs were carried out. The resulting docking poses were analyzed using AutoDockTools and Discovery Studio Visualizer 2021 Client. To validate the docking procedure, the co-crystalized ligand was previously docked to the protein, obtaining a RMSD value of ≤2 Ǻ in the redocking procedure.

### 2.6. Statistical Analysis

Data are presented as means ± SEM of measurements made on 6 animals in each group. Comparisons across three or more treatments were made using one-way ANOVA with Tukey’s post hoc test or repeated measures two-way ANOVA with Bonferroni’s post hoc test, when appropriate. All data were analyzed using the Prism 5.01 computer software (GraphPad, San Diego, CA, USA). Statistical differences were considered to be significant at *p* < 0.05.

## 3. Results

### 3.1. Synthesis

The intermediate 2-cyano-N-phenylacetamide (JM-01) (previously synthesized and characterized by Silva (2018) was used to synthesize the phenylacrylamide derivative 2-cyano-N-3-diphenylacrylamide (JMPR-01). The final molecule was obtained through the *Knoevenagel* condensation reaction presented in Figure 1. The *Knoevenagel* condensation is an important organic reaction in the formation of a carbon–carbon double bond between the carbonyl function and activated methylene groups. Many α, β-unsaturated products obtained by this type of condensation have been widely used as intermediates in the synthesis of drugs, chemicals, cosmetics, foodstuffs, and agrochemicals [17,18,19].

Due to its aprotic nature which favors catalytic activity at higher temperatures, toluene was used as a reaction medium. The catalyst used was triethylamine (NEt3), whose function in the reaction medium was to abstract a carbon proton directly linked to nitrile and carbonyl, which, due to a negative mesomeric effect, is deficient in electrons acquiring a partially positive charge. The electronic effect exerted by the ligands makes the hydrogen more acidic, facilitating the loss of the proton to form the carbanion. *Knovenagel* condensation results from the nucleophilic attack of the carbanion on the carbonyl group of the aldehyde, due to its electron-deficient nature. Protonated triethylamine transfers its proton to the oxygen of the aldehyde, forms a hydroxyl, and restores the base.

The reaction’s completion is characterized by protonation of the hydroxyl and its full exit in the form of water. This process allows the carbon to oxidize and results in the formation of a double bond. After synthesis of JMPR-01, spectroscopic and spectrometric methods were performed for its structural elucidation and determination of physicochemical parameters: theoretical partition coefficient (LogP), molar mass (MM) (Chemdraw), retention factor (Rf), and yield, shown in Table 1.

### 3.2. Structural Elucidation

Through ^1^H NMR analysis, it was possible to identify a singlet referring to the NH group of the amide that were detected with displacement at δ 10.43 ppm. This signal is outside the characteristic range of 5.0 to 9.0 ppm due to the anisotropic effect, which leaves the H less shielded. It was observed peaks characteristic of singlet (s) at δ 8.30 ppm indicative of condensation between the aldehyde and methylene group, culminating in the formation of the vinyl group (C=CH). Other aromatic hydrogens could be found with displacements ranging between δ 7.12 to 8.05 in triplet (t) and multiplet (m) form. The ^13^C NMR spectrum also confirmed the structure by the presence of signals correspondent to the compound. Two signals were observed in the negative region at 160.97 and 138.68 ppm, corresponding respectively to the carbonyl carbon and the quaternary carbon, and indicating *Knoevenagel* condensation. Likewise, it was possible to detect signals at 151.27 ppm, indicative of vinyl carbons, and at 116.68 ppm, which is diagnostic of the nitrile function. Infrared results also assisted in the structural characterization of the JMPR-01. The following bands were observed: 1682, 2227, and 3017 cm^−1^, characteristic of elongation of the carbonyl, nitrile, and amide groups, respectively. Ultimately, mass spectrometry (MS) was useful to confirm the structure of the novel synthesized compound, exhibiting the result of *m/z* = 249.10 (Spectroscopic and spectrometric data can be found in the Appendix A).

### 3.3. Thermal Characterization (DSC and TGA)

The DSC curve for JMPR-01 presents a well-defined endothermic peak, characteristic of melting, occurring at a temperature of 200.26 °C (ΔH = 222.5 J g^−1^) and a purity of 99.82%. Regarding the thermogravimetric curve, we observed a single stage of degradation between 174.35–319.92 °C with a mass loss of 96.74%. At the end of the analysis, the presence of inorganic residues (0.3545%) was verified which did not entirely decompose until 900 °C (The DSC and TGA curves are found in the Appendix A).

## 4. Biological Activity

### 4.1. In Vitro Tests

The anti-inflammatory and immunomodulatory potential of JMPR-01 was explored in a series of cell culture assays using J774 murine macrophages. The first test performed was cell viability using the Alamarblue method, which indicates impairment of cell metabolism through color change by mitochondrial enzymes [20]. For this test, JMPR-01 was used at concentrations of 6.25, 12.5, 25, 50, and 100 μM, as shown in Figure 1. Based on the results, induction of cytotoxic effects occurred at a concentration of 100 µM. Determining the cytotoxic concentration is important, since it allows targeting concentrations of JMPR-01 for use in other tests, such as dosing for nitrite and cytokines. For these tests, concentrations of up to 50 μM were tested, with no changes in cell viability after 72 h.

Macrophages are a principal cell of the immune system, which, when activated by microbial products (such as LPS and IFN-γ), produce mediators such as nitric oxide, proteolytic enzymes, and inflammatory cytokines, among others. These substances are part of the pathophysiological inflammation process promoted in tissue and vascular injury [21]. Nitrite is a decomposition product of nitric oxide, important in the cell oxidation process, and quantified in cultures of stimulated macrophages. In Figure 2, JMPR-01 at concentrations of 50–3.125 μM promoted a significant reduction (*p* < 0.05) in nitrite production compared to the un-stimulated control group. Dexamethasone (40 μM) was used as a standard.

Nitric oxide (NO) is a gas molecule which, when crossing biological membranes, produces reactive species through association with oxygen and superoxides. Its production involves nitric oxide synthase (NOS) enzymes, which are presented in three isoforms: neuronal (nNOS), endothelial (eNOS), and inducible (iNOS), using L-arginine, NADPH, and oxygen as substrates. The inducible form (iNOS) is expressed in the highest concentration in response to inflammatory stress, producing higher concentrations of nitric oxide. In macrophages, iNOs can be induced by LPS, a microbial product. NO promotes the release of leukocytes, macrophages, mast cells, endothelial cells, and platelets. NO also promotes blood flow modulation, leukocyte adhesion to vascular endothelium, and the activity of numerous enzymes which impact inflammatory responses [22,23].

The capacity of JMPR-01 to suppress production of cytokines expressed by macrophage stimulation (interleukin 1 beta (IL-1β) and tumor necrosis factor alpha (TNF-α)) was also evaluated. Based on the results, there was a significant reduction (*p* < 0.05) of TNF-α after the treatment of cultures with JMPR-01 at concentrations of 50 and 25 μM (Figure 3A). For IL-1β, JMPR-01 inhibited TNF-α at all concentrations (50 to 3.125 μM) (Figure 3B), which demonstrates its pharmacological potential even at its lowest tested concentration. Dexamethasone at a concentration of 40 μM was used as the standard drug for this assay as well.

From the dose-response curve of JMPR-01, standardized by TNF-α inhibition in J774 macrophage cultures, the CC50, EC50 and Selectivity Index (S.I.) values were obtained. The cytotoxic concentration that reduced cell viability by 50% compared to the control group was 977.25 ± 655.36 µM. The concentration of the compound for which 50% of the effect was observed was 7.02 ± 4.24 µM. From the ratio (CC50/EC50) was determined (S.I.), which showed a value of 139.20 (Appendix A. Given these results, JMPR-01 proves to be a strong candidate as an anti-inflammatory agent, encouraging subsequent in vivo assays.

Based on the observed reduced expression of TNF-α, IL-1β, and nitrite in vitro, it can be inferred that a regulatory mechanism is correlated with this transcriptional pathway effect, including a possible decrease in oxidative stress, and inhibition of key regulators of the NF-κB pathway. There are few studies evaluating the performance of phenylacrylamide derivatives in this effect, but they do suggest that their Michael acceptor regions bind to cysteine residues in Keap-1 cytosolic proteins, which act as redox sensors and negative regulators of Nrf2 (nuclear factor erythhroid 2 related factor 2), activating the expression of antioxidant and anti-inflammatory enzymes [24,25,26]. Our results suggest possible mechanisms of action of phenylacrylamide derivatives towards the inhibition of inflammatory mediators involved in inflammatory signaling, and allow for the future development of promising drugs to treat inflammatory diseases.

### 4.2. In Vivo Tests

#### 4.2.1. Peritonitis Induced Zymosan

Leukocyte recruitment involves intercellular communication between endothelium and defense cells, forming the leukocyte adhesion cascade. Subsequent phases involve the release of inflammatory mediators into the vessel lumen, activation of resident cells, and leukocyte chemotaxis towards the inflammation site [27,28]. An in vivo peritonitis assay was used to investigate the anti-inflammatory pharmacology and potential of the test drug to decrease leukocyte migration.

Zymosan (obtained from the polysaccharide wall of the fungus *Saccharomyces cerevisiae*) was used as a phlogiston agent [29]. Its action occurs through activation of components of the inflammatory machinery, such as complement system proteins, prostaglandins, leukotrienes, ROS, and platelet aggregation factor [30]. Through direct macrophage stimulation mediated by Toll-like TLR-2 and TLR6 receptors, it was observed that Zymosan was able to induce phagocytosis and the production of inflammatory mediators such as TNF-α, IL-1β, by activating the NF-κB pathway [31].

As can be seen in Figure 4 of the peritonitis model, JMPR-01 significantly (*p* < 0.05) inhibited leukocyte migration by 61.8, 68.5, and 90.5%, at respective doses of 5, 10, and 50 mg/kg. For the group treated with the standard indomethacin (10 mg/kg), inhibition was 45%. All groups were compared to control (DMSO 5% + saline).

Studies involving Zymosan-induced peritonitis reveal that in the initial phase of administration, animals present vascular permeability and increased levels of myeloperoxidase, demonstrating massive activity involving recruitment and activation of leukocytes [31,32]. In the inflammatory process, leukocyte chemotaxis involves marginalization, capture, rollover, activation, adhesion, and transmigration to the inflammation center [33]. Depending on the microenvironment where inflammation occurs, these processes include different signaling pathways and chemo-attractants [34]. These data corroborate in vitro human umbilical vein endothelial cell studies, where it was observed that IL1-β and TNF-α mediate eosinophil adhesion and trans-endothelial migration, stimulating the release of ICAM-1 (intercellular adhesion molecule-1) E-selectin, and VCAM-1 (vascular cell adhesion molecule-1) [35].

#### 4.2.2. Paw Edema

Vascular responses are the first changes seen in the inflammatory cascade. They involve vasodilation, hyperemia, and increased vascular permeability, to enable the flow of proteins, electrolytes, and exudate into the interstitium [36]. Vascular responses are mediated directly or indirectly by chemical factors caused by inflammatory stimulus in plasma cells and proteins, resulting in what can be seen clinically as edema [37].

Complete Freund’s Adjuvant (CFA) consists of inactivated *Mycobacterium tuberculosis* and mineral oil. It is an inducer of inflammatory processes, acting through mobilization and activation of antigen-presenting cells (APCs), and increasing the response of T cells [38]. In addition to immunological manifestations, CFA can trigger the release of inflammatory mediators such as PGE2, NO, leukotriene B4 (LTB4), TNF-α, IL-2, and IL-17 [39].

According to Ben et al. [40] and Gris et al. [41] in models of rheumatoid arthritis and nociception, CFA is commonly used [38,42] in experimental inflammation models as an inducing agent, having shown that it induces a chronic inflammatory response after intraplantar administration edema with a peak within 24 h occurs.

Figure 5 presents the results of evaluation of JMPR-01 anti-inflammatory activity in the CFA-induced paw edema model. Pre-treatment carried out with JMPR-01 at a dose of 50 mg/kg inhibited formation of edema from 6 h onwards, with activity ceasing by 24 h. Using the compound in a higher dose (100 mg/kg), a satisfactory inhibition was observed from 2 to 6 h, with similar activity to the Dexamethasone control, yet without being significantly maintained after 24 h.

In vitro, JMPR-01 was shown to suppress TNF-α and IL-1β production by macrophages stimulated with LPS and IFN-γ; making it plausible that this effect may contribute to the anti-edema activity obtained. This assumption is based on modulation (by TNF-α and IL-1β) of gene expression responsible for encoding the iNOS enzyme, through the p38 MAPK pathway and activation of the nuclear factor kappa-B (NF-κB) [42,43], as well as the significant reduction in nitrite production (a metabolite used to measure NO in vitro). Once expressed, iNOS catalyzes synthesis of NO, (a molecule with a stimulating action on the soluble guanylate cyclase enzyme (sGC) with consequent formation of cyclic guanosine monophosphate (cGMP)—especially in this context, due to its vasodilator activity) [44]. With the *Zymosan*-induced paw edema test being nonspecific, the action of JMPR-01 may result from inhibition of many mediators that contribute to edema promoting activity. These in silico results may contribute to predict its behavior in important targets for anti-inflammatory activity.

### 4.3. Docking Studies

In silico studies use molecular docking to predict the most favorable conformation, and orientation or “pose” assumed by a ligand, as well as the interactions established between molecules and biological targets. Consequently, molecular docking allows elucidation of fundamental biochemical processes through characterization of the behavior of small molecules when inserted in the active site [45,46].

Free binding energy (ΔG) estimates the binding affinity of the ligand–target complex and is obtained through the Gibbs–Helmholtz equation. Lower free binding energy values indicate greater stability for the complex, favoring interaction of the ligand in the active site when it assumes the particular pose [47]. The results obtained through in silico evaluation of relevant molecular targets and delineation of possible mechanisms related to the anti-inflammatory and immunomodulatory potential of JMPR-01 are described in Table 2 and Figure 6. Validation of the selected targets occurred during redocking of the co-crystallized ligands, presenting RMSD values ≤ 2.0 Å [48].

According to Table 2, JMPR-01 presents greater affinity than the co-crystallized inhibitors to the inducible nitric oxide synthase (iNOS) and phosphodiesterase 4B (PDE4B) targets. The iNOS isoform (with crucial importance to the inflammatory process) is regulated at the transcriptional level through stimulation of immune cells by pro-inflammatory cytokines or endotoxins. The regulation of iNOS isoform activity is important to maintain physiological responses and control the deleterious effects mediated by NO [49]; thus, inhibition is part of clinical management in many inflammatory diseases. This means that iNOS inhibition, as a result of the suppression of cytokines, chemokines, and adhesion molecules, in addition to the maintenance of vascular tone and permeability, (as suggested in the present paw edema model in this study, and as well as the affinity of the tested compound for iNOS) is related to the observed in vitro modulation of nitrite production [37,50].

Figure 7 presents JMPR-01 and the AR-C95791 ligand—iNOS target interactions; it is noteworthy that JMPR-01 established additional interactions with the amino acid residues Gln 263 and Glu 377, justifying greater affinity with the active site. According to Garcin et al. (2008), [51] these residues are essential for the inhibitory activity of this target, since substitutions resulted in binding affinity reductions (Kd), especially in relation to residue Glu 377, whose Kd ranged from 0.4 μM to above 100 µM. In addition to having interactions with residues in common (Val 352 and Pro 350) with the co-crystallized target inhibitor IC_50_ = 0.35 μM—iNOS isoenzyme [51], the pose established parallel to the Heme group (Figure 6A) also favored interaction with the cofactor (Hem 901).

Delineation of the affinity of JMPR-01 with PDE4 (an isoform that has specific hydrolyzing capacity on cAMP—a second messenger with modulatory action on effector cells in the pathogenesis of inflammation) was also tested [52]. PDE4 as an inhibited pharmacological target is correlated with a consequent intracellular accumulation of cAMP, which can directly activate protein kinase A (PKA) (responsible for phosphorylation of cAMP-responsive protein (CREB) and activating factor of transcription (ATF-1)), leading to increased synthesis of anti-inflammatory cytokines [53].

Roflumilast is a potent selective PDE4 inhibitor, presenting IC_50_ = 0.8 nM against PDE4 expressed in human neutrophils [54]. Figure 8 presents roflumilast and JMPR-01 with the PDE4B target. Certain shared interactions with amino acid residues (Ile 410, Asn 395, Asp 392, and His 234) are highlighted. JMPR-01 interacted in a complementary way with Ser 394 and Pro 396, as well as presented higher quality interactions at the active site, through hydrogen bonds and shorter distance charge transfer interactions, resulting in a higher interactive force and lower free binding energy values as compared to roflumilast.

The influence of COX-2 in anti-inflammatory therapy is suggested, which highlights its specific up-regulation at the site of inflammation, mainly in response to the local release of cytokines such as IL-1 and TNF-α, activators of the NF-kB transcription factor [55]. Through the metabolism of arachidonic acid, these generate pro-inflammatory and vasoactive eicosanoids such as prostaglandins (PGD2, PGE2, PGF2α), prostacyclins (PGI2), and thromboxanes (TXA2) [56]. Thus, its selective inhibition is an effective way to obtain anti-inflammatory, analgesic, and antipyretic activity without harming the physiological actions inherent to COX-1, a target inhibited by traditional NSAIDs [57].

The selective inhibitor Celecoxib, (IC_50_ = 0.132 ± 0.005 μM against COX-2), establishes interactions with amino acid residues related to selective inhibition of the enzyme, such as Ser 339, Arg 499, Phe 504 and Val 509, as observed in Figure 9 [58]. JMPR-01, being directed to the more selective region (Val 509), also shares interactions with residues Trp 373, Met 508, and Leu 370, as well as with Ala 502 and Tyr 371. Therefore, JMPR-01 demonstrated inhibition potential that corroborates the results obtained in the in vivo assays, where leukocyte migration and edema formation are significantly induced by PGE2, together with the chemotactic and vasoactive action of leukotrienes [56].

Arachidonic acid is also a 5-LOX substrate, and characterized as a crucial point in synthesis of leukotrienes and lipoxins, which are, respectively, (initiation and resolution) considered regulators of inflammation through association with the 5-LO activating protein (FLAP), the metabolite is 5-hydroperoxyeicosatetraenoic acid (5-HPETE) and, in a subsequent dehydration reaction, the leukotrienes A4 (LTA4). These, in turn, are substrates for the enzyme LTC4 synthase, responsible for the biosynthesis of cystenyl-leukotrienes (CisLT) and for LTA4H in catalysis of the final step in LTB4 biosynthesis, which is a recognized lipid chemotactic mediator and leukocyte activator [59,60].

Further, the action of JMPR-01 in 5-LOX and LTA4 hydrolase enhances its inhibitory effect on the synthesis of leukotrienes and consequent reduction in leukocyte migration, which was indicated in the zymosan induced peritonitis model. According to Sailer et al. (1996) [61], the AKBA ligand co-crystallized to the target 5-LOX presents an IC_50_ = 1.5 μM in relation to the enzyme expressed by peritoneal polymorphonuclear leukocytes. The active AKBA site is allosteric to the catalytic portion, where it binds to residues His 130, Arg 101, and Thr 137 through hydrogen bonds, shown in Figure 10 [62].

JMPR-01 interacted with two highlighted residues (His 130 and Arg 101) and also shared interactions with Val 110 and Arg 138. Further, the complementary portion composed of Glu 134, Ala 388, and Pro 98 was indicated. Yet the compound under study presented greater affinity for the enzyme LTA4 hydrolase, with a free binding energy close to the co-crystallized ligand, Bestatin, which is a reversible inhibitor of this enzyme with IC_50_ = 4.0 ± 0.8 μM [63]. In Figure 11, an interactive overlap suggests influence on the catalytic domain of the target (Zn ^2+^ coordinated by His 295, His 299, and Glu 318) from interactions with His 295, and Zn2+ that are essential for LTB4 biosynthesis of [64].

Through in silico studies, carried out in molecular docking, it was possible to predict the affinity and behavior of JMPR-01 in the catalytic sites of important inflammation targets, with satisfactory results in the enzymes: iNOs, PDE 4B and LTA4H. Inhibition of these targets suggests possible molecular mechanisms for further investigation in more specific in vitro and in vivo assays.

## 5. Discussion

The compound under study is a synthetic phenylacrylamide: 2-cyano-N-3-diphenylacrylamide derivative (JMPR-01), obtained by the *Knoevenagel* condensation reaction, structurally elucidated by ^1^H and ^13^C NMR techniques, in addition to infrared (IR) and mass spectrometry techniques, showing signs that characterized their main substituent groups, namely, the amide group, the benzene ring systems and the cyano group. It is noteworthy that phenylacrylamide derivatives have already been reported in the literature showing a promising result, affirming the molecule as a good drug candidate. Existing studies include the evaluation of the cytotoxic activity of 2-phenylacrylamide derivatives [65] and the evaluation of the anti-inflammatory activity of cyano-phenylacrylamide hybrid derived from indomethacin and paracetamol [8].

The inflammatory process is mediated by multiple agents. Experimental evaluation in inflammation models is necessary for a good understanding at the molecular level of the substance’s action. Cytotoxicity assays performed demonstrated that the compound JMPR-1 initiated the induction of cytotoxic effects at a concentration of 100 µM, without significant changes up to a period of 72 h. In addition to the assessment of cytotoxicity and cell viability alteration, the study compound at concentrations of 50–3.125 μM promoted a significant reduction (*p* < 0.05) in nitrite production compared to the unstimulated control group, and also inhibited the TNF-α at all concentrations (50 to 3.125 μM), thus demonstrating a possible modulation of nitric oxide synthases and reduced expression of TNF-α.

TNF-α is a pleiotropic cytokine, inducing the production of IL-1 and IL-6. Together, these factors promote cellular processes such as chemotaxis, increased vascular permeability and hyperalgesia. IL-1β is directly related to increased excitability and sensitization of nociceptive endings, in addition to inducing apoptosis in the responses of inflammatory and carcinogenic processes [66]. These cytokines, as well as endotoxins, such as lipopolysaccharides (LPS), promote the activation of the endothelium cells through the Toll-like receptor 4 (TLR4) and the RIG-I pathway, which promotes cytokine secretion and the expression of molecules of adhesion, such as E-selectin, VCAM-1 and ICAM-1 to facilitate leukocyte extravasation. Tissue damage arises with increased leukocyte migration due to the secretion of large amounts of inflammatory mediators and reactive molecules, which aggravate the inflammatory process [67].

Cytokines are essential mediators for the development of autoimmune and inflammatory pathologies, a result of the induction of the TNF-α signal, including the activation of NF-κB, ERK, p38 MAPK and JNK, which are gene activation pathways for IL-1β, IL-8, IL-6, chemokines, adhesion molecules and immunological enzymes, such as iNOs itself [68,69]. TNF-α, as well as IL-1, influence the activity of the signaling pathway for the NF-κB transcription factor, which regulates the gene expression of inflammatory mediators (these cytokines are overexpressed when this signaling pathway is activated by stress oxidative). This family of cytokines is composed of structurally related members, including NF-κB1 (p50/p105), NF-κB2 (dp52/p100), RelA (p65), RelB and c-Rel that are normally inactivated in the cytoplasm by inhibitory IκB proteins, mainly IκBα [70,71].

These cytokines, in turn, act on both canonical and non-canonical NF-κB activation pathways, activating the IκB kinase complex (IKK), which then mediates phosphorylation, ubiquitination and degradation of IκB and free NF-κB dimer translocation to the nucleus to initiate gene transcription (canonical pathway). The non-canonical pathway involves phosphorylation of the inhibitory domains of p100 and p105 and also results in the release of the NF-κB dimer. Activation of this signaling pathway is important, as it results in the expression of several inflammatory mediators, including many enzymes, such as inducible nitric oxide synthase (iNOS), cyclooxygenases and lipoxygenases that exert pro-inflammatory effects [72,73].

The action of the compound JMPR-1 as a modulator of nitric oxide (NO) synthases was also visualized through Molecular Docking simulations, since the score obtained by the test compound was more negative than that obtained by the co-crystallized ligand; besides that, it also established important interactions with the amino acid residues Gln 263 and Glu 377 that are essential for the inhibitory activity of this target.

The regulation of the activity of iNOS isoforms is important to maintain the physiological responses and control the deleterious effects mediated by NO [49]; thus, inhibition is part of the clinical management in many inflammatory diseases. Therefore, the inhibition of iNOS, as a result of the suppression of cytokines, chemokines and adhesion molecules, in addition to the maintenance of vascular tone and permeability, (as suggested in the present paw edema model in this study, and also the affinity of the compound tested for iNOS) is related to the observed in vitro modulation of nitrite production [37,74].

Based on the observed reduced expression of TNF-α, IL-1β, and nitrite in vitro, it can be inferred that a regulatory mechanism is correlated with this transcriptional pathway effect, including a possible decrease in oxidative stress, and inhibition of key regulators of the NF-κB pathway. There are few studies evaluating the performance of phenylacryla-mide derivatives in this effect, but they do suggest that their Michael acceptor regions bind to cysteine residues in Keap-1 cytosolic proteins, which act as redox sensors and negative regulators of Nrf2 (nuclear factor erythhroid 2 related factor 2), activating the expression of antioxidant and anti-inflammatory enzymes [23,24,25].

In evaluating the anti-inflammatory activity in in vivo models, JMPR-01 significantly inhibited (*p* < 0.05) the migration of leukocytes by 61.8, 68.5 and 90.5%, at the respective doses of 5, 10 and 50 mg/kg, assuming a better performance when compared to the control, which corresponded to indomethacin. Similarly, in the in situ inflammation model, the paw edema that JMPR-01 at a dose of 50 mg/kg inhibited the formation of edema from 6 h onwards, with cessation of activity within 24 h, could be visualized. Using the compound at a higher dose (100 mg/kg), satisfactory inhibition was observed from 2 to 6 h, with activity similar to that of Dexamethasone control, but not significantly maintained after 24 h.

Inflammatory mediators directly involved in the recruitment of leukocytes have been widely studied with the aim of elucidating their mechanisms of action. Despite a well elucidated chemotactic activity [75], Tager et al. [76] identified the involvement of Leukotriene B4 (LTB4) in the recruitment of CD-4 + T lymphocytes through the BLT-1 receptor and demonstrated that LTB4/BLT-1 T cell-mediated chemotactic activity compared to CXCL12 is one of the most effective T cell chemoattractants. An in vitro complementary study also revealed that CD-8+ T cells, after stimulation by IL-2, acquired chemotactic sensitivity to LTB4 (when released by previously activated mast cells) [77].

Cytokines such as TNF-α, IL-1, and IL-4 also play important roles in chemotaxis, acting through the up-regulation of vascular adhesion molecules, specifically E-selectin, P-selectin, and the cell adhesion molecule (ICAM -1), promoting fixation and rolling of leukocytes inside the vessel [78]. These data corroborate in vitro human umbilical vein endothelial cell studies, where it was observed that IL1-β and TNF-α mediate eosinophil adhesion and trans endothelial migration, thereby stimulating the release of ICAM-1 (intercellular adhesion molecule-1) E-selectin, and VCAM-1 (vascular cell adhesion molecule-1) [34].

The reduction in cytokine production disfavors COX-2 transcription through the NFκB pathway. In the edematogenic activity, COX-2 transcription stands out for the biosynthesis of eicosanoids, including prostaglandin E1 (PGE1), prostaglandin E2 (PGE2) and prostacyclin I2 [56]. Together, they participate in the vascular phase of inflammation, promoting vasodilation and increased permeability of the venule with consequent plasma exudation [36,56]. These, in particular PGE1 and PGE2, are capable of sensitizing afferent nociceptive nerve terminals to endogenous pain mediators, such as bradykinin, histamine and serotonin, which are intrinsically related to hyperalgesia [79].

Dexamethasone was chosen as the standard for in vitro tests of nitrite and cytokine dosages, as well as in paw edema, due to its good performance in anti-inflammatory and immunomodulatory therapy. Its mechanism involves the action on nuclear receptors involved in the transcription of anti-inflammatory mediators [80], among which we highlight those of protein origin, such as Annexin A1, known as a potent multifunctional anti-inflammatory mediator. It is known that the activity of Annexin A1 comes mainly from its active grouping called N-terminal peptide Ac2-26, which binds to the formyl-peptide receptor (FPR) grouping, promoting reduction in the production of the storm of pro-inflammatory cytokines [81]. The direct correlation between glucocorticoids and Annexin A1 can be described in in vivo studies using mice deficient in Annexin A1, where it was possible to observe an absence of response to administered glucocorticoids [82,83].

Studies by Pupjalis et al., 2011 [84], using apoptotic immune cells, demonstrated that dexamethasone-mediated Annexin A1 expression contributed to TNF-α suppression by a mechanism involving formyl peptide receptors (FPR). These data corroborate in vitro studies by Gobbetti and Cooray, 2016 [85], involving fibroblasts present in chronic inflammatory lung diseases, where it was possible to identify that mimicking the actions of Annexin A1 may have positive therapeutic effects in these pathologies, through the regulation of the NFκ-B pathway was induced by TNF-α, a mediator that had its release suppressed by JMPR-01 in in vitro studies. The suppressive activity of TNF-α presented by JMPR-01 opens doors for further investigations addressing the possible role of the same in increasing the expression of Annexin A1 or in promoting activities similar to the protein itself.

This action in the reduction of mediators can be justified by Molecular Docking, in which the compound JMPR-1 established important interactions, proving a possible action in the pathways of COX-2, 5-Lox and leukotrienes, which can be useful for conducting studies futures, as well as proof of activities that cannot be experimentally validated. Our results suggest possible mechanisms of action of phenylacrylamide derivatives towards the inhibition of inflammatory mediators involved in inflammatory signaling, and allow for future development of promising drugs to treat inflammatory diseases.

## 6. Conclusions

In conclusion, this study presents for the first time the anti-inflammatory potential of a new phenylacrylamide derivative obtained through bioisosteric ring modification of a previously studied drug. Using in vitro and in vivo approaches, immunomodulatory and anti-inflammatory properties were demonstrated, suggesting that its mechanism (in part) may be related to decreased expression of inflammatory cytokines (TNF-α and IL-1β) and nitric oxide. In the in silico studies, satisfactory couplings were observed for targets such as iNOs, COX-2 and PDE-IV, suggesting potentially multi-target interactions involving the inflammatory response. This can be further investigated. These preliminary results demonstrate the therapeutic potential of JMPR-01, making it a potential drug for inflammatory conditions, with both viable synthesis and low production costs.

## Data Availability

Data can be requested by contacting the corresponding author.

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
