# Peer review of "The Compound (E)-2-Cyano-N,3-diphenylacrylamide (JMPR-01): A Potential Drug for Treatment of Inflammatory Diseases"

_pharmaceutics, 2022, doi:10.3390/pharmaceutics14010188_

Round 1
Reviewer 1 Report
In the manuscript „the Compound (E)-2-Cyano-N,3-diphenylacrylamide (JMPR-01), a potential drug for treatment of inflammatory diseases “ the authors aim to enlighten the reader of the potential anti-inflammatory effect of the newly synthesised compound JMPR-01. The authors provided in vitro, in situ, and in vivo data showing the anti-inflammatory role of JMPR-01. The balance of pro- and anti-inflammatory responses is crucial for shaping tissue repair and resolution. Unresolved inflammation contributes and determines disease progression and pathogenesis. Thus, drugs that act anti-inflammatory can be widely used in a variety of medical conditions and the here presented drug JMPR-01 might be a promising drug candidate to circumvent harmful hyperinflammatory response.
While the provided data showed a putative anti-inflammatory effect, the authors should provide additional data to substantiate their findings. The authors need to address the following points:
- With regards to the potential therapeutic approach of JMPR-01, the authors should provide CC50, EC50 (EC90) values and the selectivity index. Please validate the anti-inflammatory EC values by determining the effect of JMPR-01 on TNFα
- Please explain why JMPR-01 excited a strong dose-dependent effect on TNFα, while higher concentration of JMPR-01 had no effect on IL-1β release.
- The in vivo data in figure 5 showed the dose-dependent effect of JMPR-01 on the leukocyte recruitment in the peritonitis model. However, data showing the effect of JMPR-01 on leukocyte adhesion in vitro are missing. Could the authors please provide additional data by performing a leukocyte adhesion assay to substantiate their finding?
- Could the authors provide experimental data for the assumption in table 2? This would verify the in situ docking studies.
- I would highly recommend a reorganisation of the result-discussion section to improve the readability of the manuscript. Please provide a result section, followed by the discussion that connect the individual data and bring them into a broader context. It might be also be better to switch the in situ data with the in vivo data to ease the readability. Here, the authors could also combine and shorten the in situ data description.
- As the authors suggest the possible therapeutic use of JMPR-01, the discussion section should include a paragraph comparing the anti-inflammatory effect of JMPR-01 with the already clinically used dexamethasone and the endogenous modulator annexin a1. To allow the clinical evaluation of the pro-resolving effect of JMPR-01, I would recommend to include the following publication for the discussion (dexamethasone: PMID: 22610172, PMID: 32956495; annexin a1: PMID: 21254404, PMID: 33378946, PMID: 27447237, PMID: 29914106, PMID: 19845684).
Minor points:
- Figure 1: correct the abbreviation GV or VG
I hope that the authors can address all my concerns adequately, and provide a revised manuscript that is suitable for publication in pharmaceutics.
Author Response
"Por favor, verifique o anexo."

Reviewer 2 Report
In this manuscript, Silva et al described a potential new drug for treating inflammation. The compound JMPR-01 is based on fragments from existing drugs indomethacin and paracetamol. They tested the compound both in vitro using murine macrophage cell lines and in vivo using edema and peritonitis models, and demonstrated that the compound can reduce production of inflammatory factors such as nitrite and cytokines in vitro and show anti-edematogenic activity and inhibitory effect on leukocyte migration. They then performed molecular docking of the compound with relevant targets such as iNOS, PDE4B and LT-A4-H, and presented potential interactions with those targets.
Overall the results are convincing that JMPR-01 has inhibitory potential for inflammatory conditions, though all quite preliminary. Only two inflammatory factors in one mouse cell line were tested in vitro, despite a vast repertoire of factors that could be affected; there was also no biophysical interactions being validated in vitro for all the molecular docking models. With limited data, it is challenging to decipher the actual molecular mechanisms of this compound in modulating inflammatory conditions, but understandable that it might be out of scope of this research.
Round 2
Reviewer 1 Report
The authors have expanded the discussion and reorganised the result section. Due to the pandemic situation they abstain from further experiments which could provide additional information. However, the manuscript is well written and the changes significantly improved the quality of the manuscript. I can recommend the acceptance of the revised manuscript.